

# Half-wormholes in SYK with one time point

**Baur Mukhametzhanov**

Institute for Advanced Study Princeton, NJ 08540, U.S.A.

## Abstract

In this note we study the SYK model with one time point, recently considered by Saad, Shenker, Stanford, and Yao. Working in a collective field description, they derived a remarkable identity: the square of the partition function with fixed couplings is well approximated by a "wormhole" saddle plus a "pair of linked half-wormholes" saddle. It explains factorization of decoupled systems. Here, we derive an explicit formula for the half-wormhole contribution. It is expressed through a hyperpfaffian of the tensor of SYK couplings. We then develop a perturbative expansion around the half-wormhole saddle. This expansion truncates at a finite order and gives the exact answer. The last term in the perturbative expansion turns out to coincide with the wormhole contribution. In this sense the wormhole saddle in this model does not need to be added separately, but instead can be viewed as a large fluctuation around the linked half-wormholes.

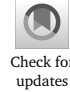

# 1   Introduction

It is becoming increasingly clear that wormholes play an important role in the physics of quantum black holes. They explain the long-time behavior of the spectral form factor [1], [2], correlation functions [3] and the entropy of Hawking radiation [4], [5]; can be traversable [6], [7], [8] and sometimes even by humans [9].

On the other hand, wormholes in AdS/CFT lead to a factorization problem [10]. From the boundary point of view, two decoupled systems $L$ and $R$ have a factorized partition function $Z = Z_L Z_R$. While from the bulk perspective, there could be contributions from wormholes connecting the two boundaries that might spoil such factorization.

Strictly speaking, there is no sharp paradox. In all known UV-complete string-theoretic examples, the wormhole solutions are either subdominant or suffer from brane-nucleation instabilities [10], [11].[1] On the other hand, in some cases, there are wormhole solutions that dominate over the disconnected solutions and have no known instabilities, but do not have a known embedding in string theory (see [11] for a nice overview of these results and references therein). In the latter case, it is not clear whether we should expect to have two decoupled boundary systems with a factorized partition function. Or if instead the boundary theory is an ensemble and its partition function does not factorize, as for example in JT gravity [2].

At the very least, wormholes contribute as off-shell configurations. So even though there is no paradox, one might wonder what is the mechanism for getting a factorized answer. A nice example, where our expectations are reasonably clear, is the spectral form factor. The wormhole, in this case called the "double-cone", describes [1] the linear ramp behavior at long times [12]. However, if we do not do either an ensemble or time averaging, the spectral form factor is expected to have large oscillations, comparable to the size of the ramp itself [13]. Therefore, we should expect that there are contributions in the bulk describing these oscillations that are comparable in size to the wormhole contribution and must be taken into account.

Recently, a toy model, where the issue can be settled, was studied by Saad, Shenker, Stanford and Yao (SSSY) [14]. They considered a finite dimensional Grassman integral that can be thought of as an SYK model where the time direction is reduced to one point. They computed a two-boundary observable $z_L z_R$ and showed that even with fixed couplings one can introduce collective field variables $G_{LR}, \Sigma_{LR}$ representing correlations between the two systems $L$ and $R$. These variables can be thought of as a proxy for the bulk description. Then the configurations with $G_{LR}, \Sigma_{LR} \neq 0$ can be viewed as wormholes.

In the theory with fixed couplings they found two types of saddles: wormhole saddle and a "pair of linked half-wormholes" saddle. Crucially, the new half-wormhole saddle is not self-averaging and depends strongly on the couplings. It disappears if we consider the average $\langle z_L z_R \rangle$. But for fixed couplings, the wormhole and linked half-wormholes combine to give a good approximation of $z_L z_R$, thus restoring factorization.

In this note, we provide some further details of this model. We derive explicit formulas for both the partition function and the half-wormhole contribution. Both are expressed as "hyperpfaffians", a generalization of pfaffian to tensors, of the tensor of SYK couplings. We then develop a perturbation theory around the linked half-wormholes. It truncates at finite order. Again, we compute every term in the expansion explicitly and show how they combine to give the exact result.

An interesting feature of the model is that the last term in the perturbation theory around linked half-wormholes coincides with the wormhole contribution. Therefore, the wormhole saddle in this model need not be added separately, but can instead be described as a large fluctuation around linked half-wormholes.

---

[1] In some of those examples, it is simply not known whether wormholes are dominant or not.

## 2 SYK with one time point

We would like to study a partition function given by a finite-dimensional Grassmann integral[2]

$$z = \int d^N \psi \, \exp\left( i^{q/2} \sum_{1 \le a_1 < \cdots < a_q \le N} J_{a_1 \ldots a_q} \psi_{a_1 \ldots a_q} \right), \qquad \psi_{a_1 \ldots a_q} \equiv \psi_{a_1} \ldots \psi_{a_q}, \qquad (1)$$

with the completely antisymmetric tensor of the couplings $J_{a_1 \ldots a_q}$ drawn out of a gaussian ensemble with zero mean and variance given by

$$\langle J_{a_1 \ldots a_q} J_{b_1 \ldots b_q} \rangle = \bar{J}^2 \delta_{a_1 b_1} \ldots \delta_{a_q b_q}, \qquad \bar{J}^2 \equiv \frac{(q-1)!}{N^{q-1}}. \qquad (2)$$

We assume that $N, q$ are both even integers and $N$ is divisible by $q$ (otherwise $z = 0$)

$$p \equiv \frac{N}{q} \in \mathbb{N}. \qquad (3)$$

In [14] this model was naturally called SYK with one time point.

In what follows, we will use capital latin letters $A, B, C, \ldots$ to denote ordered $q$-subsets of $\{1, \ldots, N\}$. For example

$$A = \{a_1 < \cdots < a_q\}, \qquad J_A \psi_A \equiv J_{a_1 \ldots a_q} \psi_{a_1 \ldots a_q}. \qquad (4)$$

We define an ordering on the $q$-subsets by their first elements[3]

$$A < B \qquad \Leftrightarrow \qquad a_1 < b_1. \qquad (5)$$

It is straightforward to compute the integral[4] (1)

$$z = \int d^N \psi \, \exp(i^{q/2} J_A \psi_A) \qquad (6)$$

$$= i^{N/2} \int d^N \psi \, \frac{(J_A \psi_A)^{N/q}}{(N/q)!} \qquad (7)$$

$$= i^{N/2} \sideset{}{'}\sum_{A_1 < \cdots < A_p} J_{A_1} \ldots J_{A_p} \int d^N \psi \, \psi_{A_1} \ldots \psi_{A_p} \qquad (8)$$

$$= \sideset{}{'}\sum_{A_1 < \cdots < A_p} \mathrm{sgn}(A) J_{A_1} \ldots J_{A_p}, \qquad (9)$$

where $\mathrm{sgn}(A) = \mathrm{sgn}(A_1 \ldots A_p)$ is the sign of the corresponding permutation. The prime on the sums means that we include only non-intersecting $q$-subsets: $A_i \cap A_j = \varnothing$, a condition enforced by the Grassmann integral. We will often use this notation below to avoid cluttering with explicit summation constraints. The resulting expression is called a "hyperpfaffian" of the tensor $J_A = J_{a_1 \ldots a_q}$ and was first introduced in [15]

$$z = \mathrm{PF} J = \sideset{}{'}\sum_{A_1 < \cdots < A_p} \mathrm{sgn}(A) J_{A_1} \ldots J_{A_p}. \qquad (10)$$

For $q = 2$ this reduces to the usual pfaffian of an antisymmetric matrix $J_{ij}$.

---

[2]We define Grassmann measure s.t. $i^{N/2} \int d^N \psi \, \psi_1 \ldots \psi_N = 1$. This will be convenient because there will be no factors of $i$ in the final expression for $z$.

[3]To be more precise, if $a_1 = b_1$ then we should compare $a_2$ and $b_2$ and so on. However, in the present work we will only be considering non-intersecting $q$-subsets $A \cap B = \varnothing$, so this issue never arises.

[4]Summation over repeated indices is implied henceforth.

## 3  Averaged theory

Ensemble averages can be computed using the hyperpfaffian expression for the partition function (10). The simplest non-trivial quantity is $\langle z^2 \rangle$

$$\langle z^2 \rangle = \langle z_L z_R \rangle = \langle (\text{PFJ})^2 \rangle \tag{11}$$

$$= {\sum_{\substack{A_1 < \cdots < A_p \\ B_1 < \cdots < B_p}}}' \text{sgn}(A)\text{sgn}(B)\langle J_{A_1} J_{B_1} \rangle \ldots \langle J_{A_p} J_{B_p} \rangle \tag{12}$$

$$= \frac{N!}{p!(q!)^p} \times \left( \bar{J}^2 \right)^p . \tag{13}$$

In the second line, we kept the only non-trivial Wick contraction.[5]  Indeed, because of the ordering both $A_1$ and $B_1$ start with the index 1. So they can be only contracted with each other. Once we contract them, the same argument goes for $A_2, B_2$ and so on. Only the diagonal terms $A_i = B_i$ survive and the result is simply the number of terms in the sum times $\left( \bar{J}^2 \right)^{N/q}$.

This, of course, is the same result as in [14], which SSSY computed from the collective field description. In particular, in the large $N$ limit $\langle z^2 \rangle$ is approximated by the wormhole saddle found in [14]

$$\text{L} \bullet\!\!-\!\!-\!\!-\!\!\bullet \text{R} \approx \text{L} \bullet\!\!-\!\!-\!\!-\!\!\bullet \text{R} \tag{14}$$

where the LHS represents the contractions of $J$'s in (12).

Next, we compute $\langle z^4 \rangle$

$$\langle z^4 \rangle = \langle z_L z_R z_{L'} z_{R'} \rangle = \sum_{\substack{n_1+n_2+n_3=N/q \\ n_i \geq 0}} \tag{15}$$

$$= (\bar{J}^2)^{2N/q} \sum_{\substack{n_1+n_2+n_3=N/q \\ n_i \geq 0}} \frac{N!}{(qn_1)!(qn_2)!(qn_3)!} \times \left[ \frac{(qn_1)!}{(q!)^{n_1} n_1!} \frac{(qn_2)!}{(q!)^{n_2} n_2!} \frac{(qn_3)!}{(q!)^{n_3} n_3!} \right]^2 \tag{16}$$

$$= \left( \frac{\bar{J}^2}{q!} \right)^{2N/q} N! \sum_{\substack{n_1+n_2+n_3=N/q \\ n_i \geq 0}} \frac{(qn_1)!(qn_2)!(qn_3)!}{(n_1! n_2! n_3!)^2} . \tag{17}$$

Each of the four hyperpfaffians corresponding to systems $L, R, L', R'$ has $p = \frac{N}{q}$ factors of $J_{A_i}$. In the most general pattern of Wick contractions, represented pictorially in (15), we split $J$'s of each system into $p = n_1 + n_2 + n_3$. One can check that $n_1, n_2, n_3$ must be the same for all four

---

[5]The prime on the sum here obviously implies only that $A$'s are non-intersecting among each other and the same for $B$'s, but no such condition between $A$'s and $B$'s. We will often abuse notation in this way to keep equations readable. Hopefully, the precise constraint will be clear from the context.

systems.[6] Then the combinatorics in (16) works as follows. First, we assign $N = qn_1 + qn_2 + qn_3$ indices to one of the three groups of $J$'s described above. There are $\frac{N!}{(qn_1)!(qn_2)!(qn_3)!}$ ways to do that. At this stage we do not count permutations within each of the three groups. Now, there are $\frac{(qn_1)!}{(q!)^{n_1}n_1!}$ ways to assign $qn_1$ indices to $n_1$ $J's$, where we do not count permutations of indices on one $J_A$ and do not count permutations of $n_1$ $J$'s. Similarly for $n_2, n_3$. Finally, assignment of indices must be the same for Wick-contracted pairs. Therefore, we have a factor of $\frac{(qn_i)!}{(q!)^{n_i}n_i!}$ for each of the six pairs of systems contracted as in (15). This agrees with the computation in collective field description in [14].

In the large $N$ limit, the three terms when one of the $n_i$'s is $N/q$ dominate, giving $\langle z^4 \rangle \approx 3\langle z^2 \rangle^2$. They correspond to the three wormhole saddles [14]

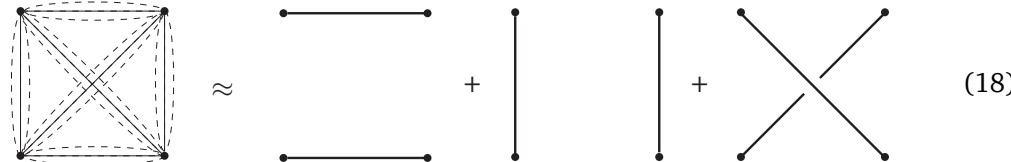

$$(18)$$

## 4   Non-averaged theory

We now turn to studying $z^2$ with fixed couplings. SSSY [14] noted that in a theory with fixed couplings we can still introduce a collective field description. This is done by inserting identity as an integral

$$1 = \int_{-\infty}^{\infty} dG \underbrace{\int_{-i\infty}^{i\infty} \frac{d\Sigma}{2\pi i/N} \exp\left[-\Sigma\left(NG - \psi_i^L \psi_i^R\right)\right]}_{\delta(G - \frac{1}{N}\psi_a^L \psi_a^R)} \exp\left\{\frac{N}{q}\left[G^q - \left(\frac{1}{N}\psi_a^L \psi_a^R\right)^q\right]\right\}. \quad (19)$$

After rotating the contour by $\Sigma = ie^{-i\pi/q}\sigma, G = e^{i\pi/q}g$, we have a representation of $z^2$

$$z^2 = \int_{-\infty}^{\infty} d\sigma \ \Psi(\sigma)\Phi(\sigma), \quad (20)$$

where the first factor

$$\Psi(\sigma) = \int_{-\infty}^{\infty} \frac{dg}{2\pi/N} \exp\left[N\left(-i\sigma g - \frac{1}{q}g^q\right)\right], \quad (21)$$

is a function that does not depend on the couplings and is highly peaked around $\sigma = 0$. While the second factor

$$\Phi(\sigma) = \int d^{2N}\psi \ \exp\left[ie^{-\frac{i\pi}{q}}\sigma\psi_a^L\psi_a^R + i^{q/2}J_A(\psi_A^L + \psi_A^R) - i^q \bar{J}^2 \psi_A^L \psi_A^R\right], \quad (22)$$

is a function that contains all of the information about the couplings. Note that $\Phi(\sigma)$ is a polynomial of order $N$ and only integer powers of $\sigma^q$ are present.

Let us summarize the findings of [14]. They showed that the integral (20) has two types of saddles. First, for $\sigma$ outside of a certain finite region near $\sigma = 0$, the function $\Phi(\sigma)$ is

---

[6]There are six lines with associated $n_i$ in (15) connecting the four corners. At each corner there is a constraint that $n_i$'s sum to $p = N/q$. Therefore, we have two independent parameters. Or, equivalently, three parameters with a constraint $n_1 + n_2 + n_3 = p$.

self-averaging and is well approximated by $\sigma^N$. In this region there are "wormhole" saddles living on the unit circle $|\sigma| = 1$ in the complex $\sigma$ plane. They reproduce the averaged answer $\langle z^2 \rangle$. Second, in a region near $\sigma = 0$ the function $\Phi(\sigma)$ is not self-averaging and has a weak dependence on $\sigma$ since it is a polynomial. Near $\sigma = 0$ it is well approximated by $\Phi(0)$. While the function $\Psi(\sigma)$ is exponentially peaked at $\sigma = 0$. This is the second type of saddle, referred to as a "pair of linked half-wormholes" in [14].[7] Therefore, SSSY concluded, in the theory with fixed couplings and at large $N$ we have an approximate identity

$$z^2 \approx \langle z^2 \rangle + \Phi(0). \tag{23}$$

This is a remarkable identity and we would like to understand it better.

## 4.1 Linked half-wormholes

We start with computing the contribution of linked half-wormholes $\Phi(0)$ (recall that $p = \frac{N}{q}$)

$$\Phi(0) = \int d^{2N}\psi \ \exp\left[i^{q/2}J_A(\psi_A^L + \psi_A^R) - i^q \bar{J}^2 \psi_A^L \psi_A^R\right] \tag{24}$$

$$= i^N \sum_{k=0}^{p} \left(-\bar{J}^2\right)^k \int d^{2N}\psi \ \frac{\left(\psi_I^L \psi_I^R\right)^k}{k!} \times \frac{\left(J_A \psi_A^L\right)^{p-k}}{(p-k)!} \times \frac{\left(J_B \psi_B^R\right)^{p-k}}{(p-k)!} \tag{25}$$

$$= i^N \sum_{k=0}^{p} \left(-\bar{J}^2\right)^k \int d^{2N}\psi \sum_{I_1 < \cdots < I_k} (\psi_{I_1}^L \ldots \psi_{I_k}^L) \times (\psi_{I_1}^R \ldots \psi_{I_k}^R) \tag{26}$$

$$\times \sum_{A_1 < \cdots < A_{p-k}} J_{A_1} \ldots J_{A_{p-k}} (\psi_{A_1}^L \ldots \psi_{A_{p-k}}^L) \tag{27}$$

$$\times \sum_{B_1 < \cdots < B_{p-k}} J_{B_1} \ldots J_{B_{p-k}} (\psi_{B_1}^R \ldots \psi_{B_{p-k}}^R) \tag{28}$$

$$= \sum_{k=0}^{p} \left(-\bar{J}^2\right)^k {\sum_{I_1 < \cdots < I_k}}' \left(\mathrm{PF} J^{(I_1,\ldots,I_k)}\right)^2. \tag{29}$$

In the second line, we kept only the terms that saturate the Grassmann integral. Next, we wrote out the sums more explicitly utilizing our convention for ordering on $q$-subsets (5). In the final line, we expressed the result in terms of the hyperpfaffian of the tensor $J_A^{(I_1,\ldots,I_k)} = J_{a_1 \ldots a_q}^{(I_1,\ldots,I_k)}$. It is defined to be the original tensor $J_A = J_{a_1 \ldots a_q}$ with indices restricted to not be in $I_1, \ldots, I_k$. This condition comes about due to the fermions in (26). The final expression can in fact be put into a more suggestive form

$$\Phi(0) = {\sum_{\substack{A_1 < \cdots < A_p \\ B_1 < \cdots < B_p}}}' \mathrm{sgn}(A)\mathrm{sgn}(B)\left(J_{A_1} J_{B_1} - \bar{J}^2 \delta_{A_1 B_1}\right) \ldots \left(J_{A_p} J_{B_p} - \bar{J}^2 \delta_{A_p B_p}\right). \tag{30}$$

This is one of our main results.

The expression (30) makes it obvious that $\langle \Phi(0) \rangle = 0$. It also clarifies how the equation (23) is satisfied. Using (10), we can write the exact answer as follows

$$z^2 = {\sum_{\substack{A_1 < \cdots < A_p \\ B_1 < \cdots < B_p}}}' \mathrm{sgn}(A)\mathrm{sgn}(B)\left(\bar{J}^2 \delta_{A_1 B_1} + \left(J_{A_1} J_{B_1} - \bar{J}^2 \delta_{A_1 B_1}\right)\right) \ldots \left(\bar{J}^2 \delta_{A_p B_p} + \left(J_{A_p} J_{B_p} - \bar{J}^2 \delta_{A_p B_p}\right)\right). \tag{31}$$

---

[7]This is contrasted with "unlinked half-wormholes", which describe $z_L$ and $z_R$ separately. In this description factorization is manifest. See [14] for details.

Now we can start expanding this by choosing either of the two terms in each of the $p$ factors. If we choose all $\bar{J}^2 \delta_{A_i B_i}$ terms, we get the wormhole contribution $\langle z^2 \rangle$ given in (12). On the other hand, if we choose $(J_{A_i} J_{B_i} - \bar{J}^2 \delta_{A_i B_i})$ we get the linked half-wormholes (30). In the section 4.3 we will see how the remaining terms are reproduced in perturbation theory around the half-wormhole.

Another way to think about linked half-wormholes $\Phi(0)$ is to note that $:J_A J_B: \equiv J_A J_B - \bar{J}^2 \delta_{AB}$ can be viewed as a normal ordered product. We can suggestively write

$$z^2 \approx \langle z^2 \rangle + :z^2:, \qquad :z^2: \equiv \Phi(0). \tag{32}$$

This of course doesn't yet explain why the remaining terms in (31) are small. We turn to this next.

## 4.2 Computation of Error

We should emphasize that the approximation (32) is not a conventional large $N$ limit. In particular, it is possible to choose a set of fixed couplings $J_A$ for which it is not a good approximation. Instead, the approximation (23) is valid for a *typical* realization of the couplings $J_A$. Namely, if we draw the couplings from the gaussian ensemble (2), we find that (32) is a reliable approximation on average. In other words, the variance (or uncertainty) of $z^2$ around $\langle z^2 \rangle + :z^2:$ is small in the ensemble (2).

To quantify the errors in (32), following SSSY we define

$$\text{Error} = z^2 - \left( \langle z^2 \rangle + \Phi(0) \right). \tag{33}$$

To show that Error is small for a typical realization of the couplings, we need to compute

$$\langle \text{Error}^2 \rangle = \langle z^4 \rangle - \langle z^2 \rangle^2 + \langle \Phi(0)^2 \rangle - 2 \langle z^2 \Phi(0) \rangle. \tag{34}$$

To calculate $\langle \Phi(0)^2 \rangle$ it is convenient to think about $\Phi(0)$ as the normal ordered $z^2$, see (32). Then the computation of $\langle \Phi(0)^2 \rangle = \langle :z^2::z^2: \rangle$ is similar to the computation of $\langle z^4 \rangle$ in (15) - (17), except that we do not include contractions between $L$ and $R$ and between $L'$ and $R'$. That is, we set $n_1 = 0$

$$\langle \Phi(0)^2 \rangle = \left( \frac{\bar{J}^2}{q!} \right)^{2N/q} N! \sum_{\substack{n_2 + n_3 = N/q \\ n_i \geq 0}} \frac{(qn_2)!(qn_3)!}{(n_2! n_3!)^2} \tag{35}$$

$$\approx 2 \langle z^2 \rangle^2. \qquad (N \to \infty) \tag{36}$$

Here, in the large $N$ limit the sum is dominated by the two terms when one of $n_i$'s is $N/q$.

To calculate $\langle z^2 \Phi(0) \rangle$ we note that because of normal ordering every $J_A$ in $\Phi(0)$ must be contracted with some $J_A$ in $z^2$. This takes up all $J$'s in $z^2$, so there are no contractions of two $J$'s in $z^2$. Therefore, we get the same result as for $\langle \Phi(0)^2 \rangle$

$$\langle z^2 \Phi(0) \rangle = \langle \Phi(0)^2 \rangle \approx 2 \langle z^2 \rangle^2. \tag{37}$$

Combining (34), (13) (17), (35), (37) we find

$$\langle \text{Error}^2 \rangle = \left( \frac{\bar{J}^2}{q!} \right)^{2N/q} N! \sum_{\substack{n_1 + n_2 + n_3 = N/q \\ 1 \leq n_1 \leq p-1 \\ n_1, n_2 \geq 0}} \frac{(qn_1)!(qn_2)!(qn_3)!}{(n_1! n_2! n_3!)^2}. \tag{38}$$

At large $N$ this sum is dominated by "boundary" terms with $(n_1, n_2, n_3)$ taking one of the four values $(1, 0, p-1), 1, p-1, 0), (p-1, 0, 1), (p-1, 1, 0)$. Altogether, we have

$$\frac{\langle \text{Error}^2 \rangle}{\langle z^4 \rangle} \approx \frac{4}{3} \frac{(q-1)!}{q} \frac{1}{N^{q-2}}. \tag{39}$$

In this sense, corrections to (23) are small for $q > 2$.

### 4.3 Perturbative expansion around linked half-wormholes to all orders

Now we turn to computing corrections to the equation (23). From the integral representation (22) of $\Phi(\sigma)$ it is clear that it is a polynomial of order $N$ and only powers of $\sigma^q$ are present

$$\Phi(\sigma) = \sum_{k=0}^{p} \frac{\sigma^{kq}}{(kq)!} \Phi^{(kq)}(0). \tag{40}$$

This is a perturbative expansion around the half-wormhole saddle $\sigma = 0$. After inserting (40) and (21) into (20), we can compute both $\sigma$ and $g$ integrals (see appendix A). The result is

$$z^2 = \int_{-\infty}^{\infty} d\sigma \ \Psi(\sigma)\Phi(\sigma) = \sum_{k=0}^{p} \frac{(-\bar{J}^2)^k}{k!} \left(\frac{i^q}{q!}\right)^k \Phi^{(kq)}(0), \tag{41}$$

where we substituted factors depending on $N, q$ for $\bar{J}^2$ in a way that will be convenient momentarily. To compute $\Phi^{(kq)}(0)$, we note that $\Phi(\sigma)$ satisfies an equation

$$\frac{i^q}{q!} \partial_\sigma^q \Phi(\sigma) = \frac{\partial}{\partial \bar{J}^2} \Phi(\sigma). \tag{42}$$

This is derived from the definition (22). Here, we think of $\bar{J}^2$ as a free parameter in (22) and set it to its value (2) after computing $\bar{J}$ derivative.[8] Now, our perturbative expansion takes the form

$$z^2 = \sum_{k=0}^{p} \frac{1}{k!} \left(\bar{J}^2\right)^k \left(-\frac{\partial}{\partial \bar{J}^2}\right)^k \Phi(0) \tag{43}$$

$$= \sum_{k=0}^{p} \Phi_k, \tag{44}$$

where we also introduced a new notation $\Phi_k$ for later convenience. If we insert $\Phi(0)$ here from (30), we see this precisely corresponds to doing the expansion of the exact answer as in (31). The operator $-\bar{J}^2 \frac{\partial}{\partial \bar{J}^2}$ acting on (30) substitutes one of the factors of $:J_{A_i}J_{B_i}:$ by $\bar{J}^2 \delta_{A_i B_i}$. So $\Phi_k$ represents a piece of (31) when we choose $\bar{J}^2 \delta_{A_i B_i}$ in $k$ of the $p$ factors and choose $:J_{A_i}J_{B_i}:$ in the rest.

Of course, the expansion (43) is just reorganizing the exact answer. What is interesting is that it has an interpretation of a semi-classical expansion at large $N$.

Again, we should emphasize that the expansion (43) is not a conventional perturbative large $N$ expansion. Instead, it is a perturbative expansion for a typical choice of the couplings in the gaussian ensemble (2). Below, we will estimate the typical values by $\Phi_k \sim \langle \Phi_k \rangle + \sqrt{\langle \Phi_k^2 \rangle}$ and show that for these typical values the expansion (43) behaves as a perturbative series at large $N$.

It is particularly interesting to consider the last term in (43)

$$\frac{1}{(N/q)!} \left(\bar{J}^2\right)^{N/q} \left(-\frac{\partial}{\partial \bar{J}^2}\right)^{N/q} \Phi(0) = \langle z^2 \rangle. \tag{45}$$

It gives the same contribution as the wormhole saddle, even though we were doing a perturbative expansion around the linked half-wormholes at $\sigma = 0$. Should we separately include the wormhole, since it is also a saddle, in addition to the linied half-wormholes with fluctuations (43)? The answer is no, because (43) already gives the exact answer. We checked this explicitly above by computing (31), (30), (43).

---

[8]In deriving (42) it is also convenient to use an identity $\frac{i^q}{q!}(\psi_a^L \psi_a^R)^q = \psi_A^L \psi_A^R$.

In this sense, the wormhole does not have to be included in our toy model path integral as a saddle. But instead, it can be viewed as a large fluctuation around the linked half-wormholes. This is possible because we could track terms of order $\sigma^N$ in the expansion around $\sigma = 0$, something that might be hard to do in more sophisticated models.

Of course, one can reverse the logic and do perturbation theory around the wormhole instead. In this case, the half-wormhole contribution $\Phi(0)$ would arise as a large fluctuation around the wormhole.[9]

One potential objection to this picture is that for a typical realization of the couplings the half-wormhole and the wormhole contributions, which represent the first and the last terms in the perturbative expansion (43), are of the same order

$$\langle z^2 \rangle \sim \sqrt{\langle \Phi(0)^2 \rangle}\,. \tag{46}$$

While the remaining terms are suppressed, as we discussed in the section 4.2. This means that when we do the expansion (43), for low orders of perturbation theory $k$, corrections become more suppressed. But at some point in the expansion they start increasing again and the last term is of the same order as the first one. This seems reminiscent of the fact that perturbation theory in higher dimensional QFTs is often only asymptotic and not convergent. Let us make this more precise in our model.

We would like to estimate $\Phi_k$ for a typical realization of the couplings. The average value vanishes $\langle \Phi_k \rangle = 0$ for $k \neq p$, because $\Phi_k$ contains at least one normal ordered factor $:J_A J_B:$. So we need the variance $\langle \Phi_k^2 \rangle$. The computation is similar to $\langle \Phi(0)^2 \rangle$ in (35) and $z^4$ in (15). The answer is simply given by the diagram (15) where we set $n_1 = k$

$$\langle \Phi_k^2 \rangle = \left( \frac{\bar{J}^2}{q!} \right)^{2N/q} N! \frac{(qk)!}{(k!)^2} \sum_{\substack{n_2+n_3=p-k \\ n_i \geq 0}} \frac{(qn_2)!(qn_3)!}{(n_2!n_3!)^2}\,. \tag{47}$$

While

$$\langle \Phi_k \Phi_{k'} \rangle = 0\,, \qquad k \neq k'\,. \tag{48}$$

In particular, this is related to Error computed in the section 4.2 (see (33), (38))

$$\text{Error} = \sum_{k=1}^{p-1} \Phi_k\,, \tag{49}$$

$$\langle \text{Error}^2 \rangle = \sum_{k=1}^{p-1} \langle \Phi_k^2 \rangle\,. \tag{50}$$

For large $N$ and fixed $k$, the sum (47) is dominated when one of the $n_i$'s is $p-k$ and we estimate

$$\frac{\langle \Phi_k^2 \rangle}{\langle z^4 \rangle} \propto \frac{1}{N^{k(q-2)}}\,, \qquad (N \to \infty,\ k \text{ - fixed})\,. \tag{51}$$

We see that for $q > 2$ higher orders $k$ of perturbation theory give supressed corrections, as long as $k$ is not too large. However, when $k$ becomes of order $N$, corrections start growing. And in the complementary regime we have contributions of the same order as (51) ($p = N/q$)

$$\frac{\langle \Phi_{p-k}^2 \rangle}{\langle z^4 \rangle} \propto \frac{1}{N^{k(q-2)}}\,, \qquad (N \to \infty,\ k \text{ - fixed})\,. \tag{52}$$

---

[9]A similar phenomena was recently observed in the tensionless string [16], where the perturbative expansion around the wormhole geometry background can be computed exactly. And it gives a factorized answer without including any other semi-classical geometries as additional saddles.

## 5 Two replicas with a coupling

Another interesting variation of the model, considered in [14], is to add a coupling $\mu$ between the two replicas

$$\zeta(\mu) = \int d^{2N}\psi \, \exp\left[\mu\psi_a^L\psi_a^R + i^{q/2}J_A(\psi_A^L + \psi_A^R)\right].\tag{53}$$

This can be thought of as either analogous to the coupling between the two boundaries in the eternal traversable wormhole [8], or as an SYK model with two instants of time. Working in the collective field description, SSSY found [14] that for $1/N \ll \mu \ll 1$ the wormhole contribution gets enhanced relative to the half-wormhole and $\zeta(\mu)$ becomes self-averaging.

In this section, we compute $\zeta(\mu)$ exactly, similarly to the section 4.3. And show how the coupling $\mu$ leads to the enhancement of the wormhole from this point of view.

The coupling $\mu$ modifies the formula (20) by a shift of the argument of $\Phi$

$$\zeta(\mu) = \int_{-\infty}^{\infty} d\sigma \, \Psi(\sigma)\Phi(\sigma - ie^{i\frac{\pi}{q}\mu}\mu).\tag{54}$$

A computation similar to (40), (41) shows (see appendix A)

$$\zeta(\mu) = \sum_{k=0}^{p} \zeta_k(\mu)\Phi_k,\tag{55}$$

$$\zeta_k(\mu) = \sum_{n=0}^{k} \frac{k!}{(k-n)!(nq)!}\left(\frac{q}{N}\right)^n (N\mu)^{nq},\tag{56}$$

where $\Phi_k$ were defined in (44).

We estimate the wormhole contribution $k = p = N/q$ at large $N$ and $1/N \ll \mu \ll 1$[10]

$$\zeta_p(\mu) \approx \sum_{n=0}^{\infty} \frac{1}{(nq)!}(N\mu)^{nq} \approx \frac{1}{q}e^{N\mu}.\tag{57}$$

While the half-wormhole and small fluctuations around it contribute

$$\zeta_k(\mu) \sim \mu^{kq}N^{k(q-1)}, \qquad (N \to \infty, \, k\text{ - fixed}).\tag{58}$$

In this case the sum (56) is dominated by the last term $n = k$.

Altogether, we find that the half-wormhole contribution is suppressed in the regime $1/N \ll \mu \ll 1$ and the partition function is dominated by the wormhole and is self-averaging

$$\zeta(\mu) \approx \frac{1}{q}e^{N\mu}\langle z^2\rangle.\tag{59}$$

In the collective field description of [14], $e^{N\mu}$ comes from the value of $e^{\mu NG}$ on the dominating wormhole saddle $G = 1$. And the factor $\frac{1}{q}$ corresponds to breaking of the degeneracy between $q$ wormhole saddles by the coupling $\mu$.

---

[10]In this regime the sum is dominated by $n \sim N\mu$.

## 6 Discussion

In this paper we considered a simple finite-dimensional SYK model with fixed couplings living at one time point. The partition function of this system is given by the hyperpfaffian of the tensor of SYK couplings. Working in the collective field description, we explained how the exact factorized answer for the square of the partition function arises in perturbation theory around the linked half-wormholes. Finally, we observed that in this model it is enough to consider one saddle point and fluctuations around it. While contributions of other saddles arise as large fluctuations.

It would be interesting to know if this simple model has a dual 1d gravity system, perhaps along the lines of [17]. And if any our findings carry over to the full-fledged SYK and more sohpisticated models of holography.

## 7 Ackhonwledgements

I am grateful to Lorenz Eberhardt, Adam Levine, Juan Maldacena, Phil Saad and Ying Zhao for useful discussions. I am supported by NSF grant PHY-1911298.

## Appendix A: Computation of $\sigma$ integrals

Here we compute the integrals over $\sigma$ that were used the main text. First, we compute

$$\int_{-\infty}^{\infty} d\sigma\ \Psi(\sigma)\sigma^{kq} = \int_{-\infty}^{\infty} \frac{dg}{2\pi/N} e^{-\frac{N}{q}g^q} \int_{-\infty}^{\infty} d\sigma\ e^{-iNg\sigma}\sigma^{kq} \tag{60}$$

$$= \int_{-\infty}^{\infty} \frac{dg}{2\pi/N} e^{-\frac{N}{q}g^q} \times \left(\frac{i}{N}\right)^{kq} \frac{2\pi}{N}\delta^{(kq)}(g) \tag{61}$$

$$= \left(\frac{i}{N}\right)^{kq}\left(-\frac{N}{q}\right)^k \frac{(kq)!}{k!} \tag{62}$$

$$= \frac{(kq)!}{k!}\left(-\frac{i^q}{q!}\bar{J}^2\right)^k. \tag{63}$$

Similarly, we have ($m \equiv i e^{i\frac{\pi}{q}}\mu$)

$$\int_{-\infty}^{\infty} d\sigma\ \Psi(\sigma)(\sigma-m)^{kq} = \int_{-\infty}^{\infty} \frac{dg}{2\pi/N} e^{-\frac{N}{q}g^q} \int_{-\infty}^{\infty} d\sigma\ e^{-iNg\sigma}(\sigma-m)^{kq} \tag{64}$$

$$= \int_{-\infty}^{\infty} \frac{dg}{2\pi/N} e^{-\frac{N}{q}g^q} \times e^{-iNmg}\left(\frac{i}{N}\right)^{kq} \frac{2\pi}{N}\delta^{(kq)}(g) \tag{65}$$

$$= \left(\frac{i}{N}\right)^{kq} \int_{-\infty}^{\infty} dg\ \delta(g)\ \partial_g^{kq}\left(e^{-\frac{N}{q}g^q - iNmg}\right) \tag{66}$$

$$= \frac{(kq)!}{k!}\left(-\frac{i^q}{q!}\bar{J}^2\right)^k \times \sum_{n=0}^{k} \frac{k!}{(k-n)!(nq)!}\left(\frac{q}{N}\right)^n (N\mu)^{nq}. \tag{67}$$



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
