# Peer review of "Half-wormholes in SYK with one time point"

_SciPost Physics, doi:SciPost Phys. 12, 029 (2022)_

## Round 3 · Referee Report · Anonymous (Referee 1) · 2021-8-29

Report

This paper studies the "wormhole" and "half-wormhole" saddles in the SYK model with one time point. The topic of the paper is interesting and the results are new and well written. Therefore, I recommend for publication in SciPost Physics once the author addresses the following points. \\

$\bullet$ There is no small coupling constant for $\Phi(\sigma)$ in (22), so it's not clear in what sense these "saddles" are dominant and fluctuations around these saddles are suppressed. It would be good to explain this point more. \\

$\bullet$ In section~4.2, the author states that the approximation (23) is valid for a typical realization of the couplings. It is needed to describe the criteria for this 'typicalnes' of the couplings. More concretely, in section~4.2, it is shown that $\langle {\rm Error} \rangle^2$ is small for large $N$, but this does not mean the "Error" itself is small. It would be good to explain for what type of configuration of the couplings, the "Error" itself is indeed small. \\

$\bullet$ The only reason that we expect the ordinary (one-dimensional) SYK model is related to AdS$_2$ gravity is the symmetry breaking pattern of $SL(2,R)$. It's desirable to clarify why this type of reduction to one time point is still meaningful for gravitational theory. \\
  • validity: -
  • significance: -
  • originality: -
  • clarity: -
  • formatting: -
  • grammar: -

Author:  Baurzhan Mukhametzhanov  on 2021-09-03  [id 1732]

(in reply to Report 1 on 2021-08-29)

I would like to thank the referee for reading the manuscript and address the raised issues:

1),2) The first two questions are related, so I'll address them together. Indeed, as explained in the beginning of section 4.2, the approximation (23) is valid only for a ''typical'' realization of the couplings. This is expressed mathematically, by computing $\langle Error^2 \rangle$ and showing it is suppressed at large $N$. This is what is meant by the couplings being ''typical''.

As pointed out by the referee, this doesn't mean that Error itself for fixed couplings is small. This point is made in the first two sentences of section 4.2. In fact, by choosing some very special set of couplings it is possible to make $z^2$ deviate substantially from the approximation (23). Nevertheless, for most couplings, or on average, the Error is small.

I agree with referee's sentiment that it would be nice to know for what type of couplings the Error itself is small, but I do not have anything to add on this point at the moment.

3) A relation between SYK with one time point and some putative 1d gravity is purely speculative. At the moment, the only reason one might think it might have anything to do with a gravity theory, is that it has features that are similar to (one-dimensional) SYK. However, having gained some intuition in the toy model, one could look for a similar mechanism of factorization in theories that are known to have bulk duals, e.g. SYK, matrix models or higher dimensional QFTs.

I can make these points clearer in the paper, if necessary.

Anonymous on 2021-10-30  [id 1890]

(in reply to Baurzhan Mukhametzhanov on 2021-09-03 [id 1732])

Sufficiently addressed the points I raised. Good for publication.

---

## Round 3 · Referee Report · Anonymous (Referee 2) · 2021-9-16

Report

This work is a future development of a previous work by Saad, Shenker, Stanford and Yao about resolving the factorization puzzle in the SYK model with one time point. The main result is the observation that the wormhole saddle can be obtained as a truncation of a perturbative expansion along the half wormhole saddle. This is explained clearly in equation 43 in the paper.

One question I had is it is not clear to me that this is perturbative expansion in 1/N, which is what usually do in a large N system. From the discussion below equation 43, it seems that it is a perturbative expansion in J^2. Maybe the author can clarify this point a little bit.

There are some minor presentation issues of the paper, such as the lack of definition of sin(A) in equation A and how to go from equation 29 to 30.
  • validity: -
  • significance: -
  • originality: -
  • clarity: -
  • formatting: -
  • grammar: -

Author:  Baurzhan Mukhametzhanov  on 2021-09-22  [id 1773]

(in reply to Report 2 on 2021-09-16)
Category:
answer to question

I would like to thank the referee for reading the manuscript and comments and address his/her questions:

1) Indeed, the expansion in eq.(43) for a given fixed set of couplings is not an expansion at large $N$ in the usual sense for any fixed set of couplings. One can happen to choose a particularly bad set of couplings for which $\Phi_k$ are not suppressed as we go to higher orders $k$.

Instead, as in the paper of Saad, Shenker, Stanford and Yao, it is an expansion at large $N$ for a ''typical'' choice of couplings. This is explained in eqs. (50), (51) (or the exact result (46)) and the discussion around them. So if one randomly chooses a fixed set of couplings, with high probability the expansion (43) would be an expansion at large $N$. So it is not a usual Taylor series at large $N$. But for most randomly chosen fixed set of couplings it is an expansion at large $N$.

2) By definition $sgn(A)$ is the sign of the permutation $A_1...A_p$, where $A_j = \left( a_1^{(j)}<...< a_q^{(j)} \right) $ is an ordered $q$-subset (see (4)).

I can add this definition as well as a clarification about eqs. (29), (30) in the new version of the draft.

---

## Round 3 · Referee Report · Anonymous (Referee 3) · 2021-9-26

Report

This paper presents an interesting result in the context of a recent work by Saad-Shenker-Stanford-Yao (SSSY). The main result is an exact computation for the partition function $z^2$ with two boundaries in the SYK model at one point in time. This takes the form of a perturbative expansion that truncates at finite order. It is shown that the wormhole can be seen as a fluctuation of the half-wormhole contribution, so that it should not be included separately. The result is significant and meets the criteria for publication in SciPost Physics. I recommend the paper for publication if the following minor points can be addressed:

  1. The terminology "half-wormhole contribution" is a bit confusing as it could equally refer to the "unlinked half-wormhole contribution" which is the RHS of (3.13) in SSSY. Then it's obvious that the wormhole should not be included (as it would contradict factorization). Of course, the present paper is about the "linked half-wormhole contribution" which appears in the LHS of (3.13). Then it's a non-trivial and interesting result that the wormhole should not be included (as it appears as a fluctuation of the linked contribution, as the author has shown). It would be great to perhaps add a few words about this.

  2. I also second another referee about the perturbative expansion (43). The fact that this is actually an expansion in 1/N (only for a typical realization) could be explained better.

---

## Round 4 · Referee Report · Anonymous (Referee 2) · 2021-10-25

Report

The author has addressed my questions, and therefore I recommend publication.

---

## Round 4 · Referee Report · Anonymous (Referee 3) · 2021-10-30

Report

All my comments have been addressed so I recommend the paper for publication.

---

## Round 4 · List of Changes

1) Change in the title from "Half-wormhole in ..." to "Half-wormholes in ...". 2) "Half-wormhole" is changed to "linked half-wormholes" in several places. 3) Definition of $sgn(A)$ added after (9) 4) Paragraphs at the beginning of section 4.2 and at the end of page 9 are added to clarify the meaning of the large $N$ expansion for typical couplings.

---

## Editorial Decision

published